# Performance of Marmoset Monkeys as Embryo Donors Is Reflected by Different Stress-Related Parameters

**DOI:** 10.3390/ani12182414

**Published:** 2022-09-14

**Authors:** Charis Drummer, Julia Münzker, Michael Heistermann, Tamara Becker, Sophie Mißbach, Rüdiger Behr

**Affiliations:** 1Platform Degenerative Diseases, German Primate Center—Leibniz Institute for Primate Research, Kellnerweg 4, 37077 Göttingen, Germany; 2DZHK, Partner Site Göttingen, 37099 Göttingen, Germany; 3Endocrinology Laboratory, German Primate Center—Leibniz Institute for Primate Research, Kellnerweg 4, 37077 Göttingen, Germany; 4Animal Husbandry, German Primate Center—Leibniz Institute for Primate Research, Kellnerweg 4, 37077 Göttingen, Germany

**Keywords:** stress, anesthetics, fertility, embryo, non-human primates, marmoset monkey

## Abstract

**Simple Summary:**

Does chronic distress from repeated anesthesia and frequent handling impair fertility and well-being in marmosets, or do the animals become used to study procedures over time? In a retrospective study, we calculated the average number of embryos isolated from early-pregnancy-stage marmoset monkeys/monthly cycle (embryo retrieval rate; ERR) in an experimental setting, including monthly anesthesia for uterus flushing and frequent (~twice/week) blood sampling in an awake state. Based on this, the monkeys were subdivided into two groups: one with high ERR and one with low ERR. ERRs were then related to markers of chronic stress (e.g., blood cortisol and weight fluctuations) in the animals. During the first year of study, the group with a higher ERR already seemed to be more stress-tolerant, as indicated by (1) low blood cortisol levels, (2) only minor body weight fluctuations, and (3) lower stress ratings in a subjective evaluation. By contrast, animals from the group with a low ERR showed signs of increased stress. Importantly, the animals from the low ERR group did not improve over time, whereas animals from the high ERR group exhibited constantly high ERRs over time. We conclude that some marmoset monkeys can handle frequent blood sampling and monthly anesthesia for embryo collection without being stressed, while others appear to be stressed from the beginning and do not improve over time. The determination of stress characteristics of animals before inclusion in future studies could reduce animal numbers and improve animal welfare.

**Abstract:**

Non-human primates (NHPs) serve as embryo donors for embryo collection in order to mimic genetic diseases in humans by genetic modification. Reproductive health of the embryo donors is crucial, and chronic distress needs to be avoided. Embryo retrieval rates (ERR), anti-Müllerian hormone (AMH) concentrations, cortisol levels, and body weight fluctuations were assessed as markers for fertility and distress. With regard to successful embryo retrievals (total *n* = 667), the animals were either used for extended periods (long-term group; LTG) or only for short periods (short-term group; STG). Retrospective evaluation expectedly showed that animals in the LTG had a higher ERR than animals in the STG (*p* < 0.0001). Importantly, ERR in the LTG remained stable throughout the experimental period, and high embryo rates were already encountered during the first year of experimental use (*p* = 0.0002). High ERR were associated with high AMH and low cortisol levels, and minimal body weight fluctuations following anesthesia, indicating a superior ability of the LTG animals to handle distress. We conclude that the long-term experimental use of marmosets does not impair their fertility or health status per se, supporting the view that animal reuse can be in accordance with the 3R-principle, implying reduction, replacement, and refinement in animal experimentation.

## 1. Introduction

Experimental animal research is conducted under the premise of the 3Rs: reduction, refinement, and replacement [1]. Reduction in animal numbers requires a study design that allows one to obtain the intended information with as few animals as possible without losing sufficient statistical power. Refinement aims at smart procedures combining the requirements of the scientific study with optimal animal welfare and housing management. Replacement aims at the implementation of alternative methods, such as cell culture models, or in silico modeling in order to avoid animal experiments.

Implementing both reduction and refinement in *one* experiment may not always be feasible. The long-term and/or repetitive use of animals within a study or in subsequent studies can clearly reduce the total number of animals; however, care must be taken to avoid excessive cumulative distress and suffering of reused animals, as this would be a violation of the “refinement” requirements. Thus, there is a conflict of goals between minimizing the number of animals (to be then used in the long term) and avoiding excessive distress on individual animals.

In this context, it also needs to be considered that, particularly with larger laboratory animals such as non-human primates (NHPs), if possible, individuals are repeatedly used to keep the number of animals used for research purposes low. Furthermore, it can be speculated that repeated use of already well-trained animals adapted to the experimental procedures may be associated with reduced distress levels compared with animals not acquainted with the experimental procedures. However, to the best of our knowledge, there are no studies published addressing this issue. Moreover, not only is reuse of relevance, but also the selection of the appropriate animals for the purpose of the study. One marker for the selection of female NHPs for studies in reproductive biology might be the determination of anti-Müllerian hormone (AMH), which is a member of the transforming growth factor beta family and is produced in the granulosa cells of preantral and antral ovarian follicles [2]. It is a marker for reproductive capacity in women [3], can predict poor response to ovarian stimulation [4], and is the preferred marker of the ovarian reserve [5]. AMH is also a marker of the ovarian reserve in cynomolgus [6] and vervet monkeys [7], and data suggest that this also applies to the marmoset monkey [8].

The genetic modification of mammals was mainly developed and established in mice, and has been applied, particularly since the intervention of CRISPR/Cas, in many animals species serving as models of human diseases. In the last few years, the genetic modification of common marmosets (*Callithrix jacchus*) [9], a small and easy-to-handle NHP species, has attracted considerable attention, and further established the use of this species in animal models in biomedical research areas, such as neurobiology [10], reproductive biology [11], and stem cell research [12]. Genetic modification requires the application of artificial reproductive technologies (ART), such as the collection of oocytes for in vitro fertilization, or the isolation of naturally fertilized embryos for the subsequent application of gene-editing tools [13], such as CRISPR/Cas [14]. Moreover, ART procedures can be performed repeatedly in marmosets, but each time requires potentially stressful general anesthesia. In this context, the maximum appropriate number of anesthesias applied to an individual animal is an important parameter for an optimized implementation of the 3R-principle.

Repetitive anesthesia may represent an accumulating strain [15]. Side effects such as hypothermia can occur [16], and in combination with a high surface area to body mass correlation, dehydration may also impair drug metabolism [17]. In detail, the metabolic impact of anesthesia is reflected by increased levels of aspartate-aminotransferase [11] and creatinine kinase (CK) in marmosets [18]. A reduced metabolism and compromised well-being may not influence only the body weight, but also the reproductive state of an animal [19], which, in the end, can hamper the outcome of long-term ART studies. Diminishing the reuse of animals would, in turn, increase the number of animals used in research and counteract “reduction” in the 3Rs. Reuse of NHPs in experimental studies does not indicate by itself an animal welfare concern, but assessment of the cumulative burden should be taken into consideration for a decision of reuse [20]. A broad range of methods is applied to laboratory animals of different species, ranging from behavioral tests to severe surgical interventions. Therefore, the development of well-defined cutoffs for maximum numbers of experiments, as well as for a specific amount of strain data over an animals’ lifetime is hardly possible. Moreover, evidence-based general criteria to make reasonable decisions which type of experimental procedures and what number of repetitions would lead to excessive cumulative distress, are scarce. To the best of our knowledge, no specific guidelines on the reuse of NHPs exist, as only general statements for animal welfare are published, for example by the National Centre for the Replacement, Refinement and Reduction of Animals in Research (NC3Rs). However, it is important to gain knowledge that allows transparent, fact-based decisions in terms of whether the re- or even long-term use of animals can be medically, legally, and ethically justified or not.

In light of this controversy, we investigated whether repeated general anesthesia in the context of preimplantation embryo collection via uterine flushing, and the associated frequent handling of female common marmosets (*Callithrix jacchus*), have negative long-term effects on health, well-being, and reproductive outcome. Specifically, we retrospectively analyzed (i) ovarian cycle stability under repeated anesthesia, (ii) liver enzymes in the blood to assess potential hepatotoxic side effects of repeated anesthesia, (iii) ERRs, and (iv) AMH serum concentrations as an indicator of fertility, as well as (v) blood cortisol and (vi) body weight changes after anesthesia to estimate the potential level of distress of animals. Our data indicate that marmosets differ in their innate distress resilience, which should be taken into consideration when designing and conducting animal research.

## 2. Materials and Methods

This retrospective study comprises data from three projects involving embryo retrieval procedures (license numbers 10/0063, 14/1652, and 14/1495), all of which were approved by the federal authority and conducted at the German Primate Center (Deutsches Primatenzentrum, DPZ)—Leibniz Institute for Primate Research, Platform Degenerative Diseases, Göttingen (Germany) between 2010 and 2019.

### 2.1. Animals and Animal Housing

All experiments were carried out in accordance with relevant guidelines and regulations. Marmoset monkeys (*Callithrix jacchus*) were obtained from the self-sustaining breeding colony of the German Primate Center. Health and well-being of animals were monitored daily by experienced veterinarians and/or animal caretakers. Documentation of body weight and clinical inspection during each procedure were conducted by NHP-experienced veterinarians. All interventions were performed by experienced veterinarians or by trained staff under the supervision of veterinarians. The numbers of animals used to generate the data for each part of the study are given in the respective Materials and Methods section as well as in the legend to each figure. An overview of the animals and sample numbers is provided in Appendix A. Closely related females were kept in different rooms to avoid reproductive suppression. In both groups, two females were sisters either from the same litter or different litters. Other females were not closely related to each other (were neither sister, mother, or grandmother).

Marmoset monkeys are social tree-living New World monkeys originating from the tropical northeast region of Brazil. Accordingly, the animals were housed as heterosexual pairs in a temperature- (25 ± 1 °C) and humidity-controlled (65 ± 5%) facility. These parameters were controlled daily. Room air was changed several times per hour and filtered adequately. Illumination was provided by daylight and additional artificial lighting on a 12.00:12.00 h light:dark cycle. Each cage, consisting of stainless steel, had a vertical orientation (165 cm (height) × 65 cm (width) × 80 cm (depth)), and was furnished with wooden branches and shelves for environmental enrichment, and a wooden sleeping box. The housing room and the cages were cleaned with water at weekly intervals. The animals were fed ad libitum with a pelleted marmoset diet (ssniff Spezialdiäten GmbH, Soest, Germany). In addition, 20 g mash per animal was served in the morning and 30 g fruits or vegetables mixed with noodles or rice were supplied in the afternoon. Solid as well as liquid *Gummi arabicum* was also provided daily for nutrition or training. Furthermore, once per week, mealworms or locusts were served in order to provide adequate nutrition. Drinking water was available at any time. All materials were changed, cleaned, and sterilized regularly.

### 2.2. Embryo Retrieval

Data were collected from 32 female adult common marmosets (age range 15–122 months) during 667 embryo retrievals under anesthesia. Retrievals with poor technical performance were excluded, all other attempts were included, even if no embryo was found in the flushing medium. The number of embryos collected ranged from 0 to 4 /flush. Importantly, an outcome of 0 embryos did not lead to the exclusion of this experimental result or of the animal from the analysis. Only the quantity of embryos served as a marker for fertility, irrespective of the developmental stage of the embryo. Some animals were temporarily excluded from experimental procedures due to temporary veterinary treatment (*n* = 8) or fully excluded from analysis after severe pathologic event with possible impact on the reproductive system (*n* = 8). Female common marmosets were used as embryo donors, and their ovarian cycles were monitored using blood progesterone measurements (see below) twice a week to determine the timing of ovulation. This allows one to schedule the optimal time point for embryo retrieval of naturally fertilized embryos under anesthesia. The total number of anesthesias applied to each animal varied from 6 to 50. Two procedures for embryo retrieval were performed as described previously [21]. Presented data were derived from minimally invasive and invasive procedures of embryo retrieval. Invasive embryo collection was performed by an abdominal incision as described in Sasaki et al. [22], except for the applied anesthesia regime, where Diazepam 10 mg/mL (0.05 mL/animal Diazepam^®^, Ratiopharm, Ulm, Germany) plus a mixture of ketamine 10% (50 mg/mL; Ketamin^®^, WDT, Garbsen, Germany), xylazine 2% (10 mg/mL; Xylazin^®^, WDT, Germany), and atropine 0.5 mg/mL (0.1 mg/mL, Atropinsulfat^®^, Eifelfango, Bad Neuenahr-Ahrweiler, Germany) was used. For analgesic medication, Meloxicam (0.1 mL Metacam^®^ i.m., Boehringer Ingelheim, Ingelheim, Germany) was administered. After surgery, the animals received antibiotic therapy (amoxicillin trihydrate; 0.1 mL i.m. Duphamox^®^, Zoetis, Berlin, Germany) for 6 days. Minimally invasive embryo collection was performed as described previously [23], under short-term anesthesia with Diazepam 10 mg/mL (0.05 mL/animal Diazepam^®^, Ratiopharm, Germany) and Alfaxalon (0.1 mL/100 g bodyweight; Alfaxan^®^, Jurox, Great Britain) combined with the same analgesic medication as for the invasive embryo retrieval. Both procedures showed the same efficiency and did not differ in their ERRs or embryo flush success rates [24]. The invasive method was applied only to a small portion of all cases (7.3%; 49/667). Its impacts on AMH, cortisol, and body weight were not analyzed separately.

### 2.3. Ovarian Cycle Monitoring

To monitor the ovarian cycle, we measured progesterone concentrations in plasma samples taken twice a week from either the arteria or vena femoralis on awake animals. Progesterone concentration was determined by a direct, non-extraction enzyme immunoassay using an antiserum raised in sheep against progesterone-11-hemisuccinate-BSA. Prior to analysis, plasma samples were diluted 1:30 with TRIS buffer (pH 7.4) to establish progesterone concentrations within the working range of the assay. The assay was then performed as described by Heistermann et al. (1993), except that progesterone-3-CMO-horseradish peroxidase was used as the enzyme conjugate [24]. Intra- and inter-assay coefficients of variation (CV), calculated by replicate measurements of high- and low-value quality controls were <10% and <15%, respectively. Progesterone concentrations were available from 4 months before the onset of anesthesia for embryo retrieval until the end of the study. Animals with progesterone concentrations <10 ng/mL were considered to be in their follicular phase [25]. At the end of the follicular phase, ovulation occurs the day before progesterone rises above 10 ng/mL, followed by the luteal phase [26]. In order to precisely schedule embryo retrieval, a synthetic prostaglandin F2alpha analog (0.2 mL of a mixture of 0.1 mL Cloprostenol (Estrumate^®^, Intervet, Unterschleißheim, Germany) and 3.2 mL Ringer lactate solution) was administered to regularly cycling female marmosets during their luteal phase in order to induce luteolysis and reinitiate a new ovarian cycle [27]. Embryo retrievals were only scheduled after Estrumate injection with successful resetting to the follicular phase. This was approved by progesterone determination from blood samples afterwards.

### 2.4. Determination of Liver Enzymes in Plasma Samples

Embryo retrievals were only performed on clinically healthy and medically untreated animals. Clinical symptoms of diminished health, such as temporary diarrhea or the occurrence of hematoma with physical disturbance, led to veterinary intervention and temporary exclusion of the animal from the experimental procedures until full recovery.

To determine whether repeated application of anesthesia with ketamine and xylazine causes liver damage, we assessed plasma concentrations of the liver enzymes alanine transaminase (ALT) and gamma-glutamyl transpeptidase (GGT) in four randomly selected female marmosets with an average age of 73 months (SD ± 18 months) during routine blood sampling. From those four animals, two were from the short-term group and two from the long-term group. These animals were included in the study for between 1 to 3 years with approximately one anesthesia per month. Liver enzymes were assessed on the day before anesthesia (as baseline level), and 3 and 6 weeks after anesthesia as a check-up for veterinary purposes. Both ALT and GGT were measured on the in-house system Dimension Xpand Plus Integrated Chemistry System (Siemens Healthcare GmbH, Erlangen, Germany).

### 2.5. Determination of Anti-Müllerian Hormone (AMH) in Plasma Samples

AMH was measured in plasma samples of a subset of 10 animals (all at the age of 3 years), randomly selected from the STG (*n* = 5) and the LTG (*n* = 5). The plasma samples were residual samples from the routine ovarian cycle monitoring of embryo donors. AMH was quantified using a commercial ultrasensitive AMH ELISA Kit (Cat. No. AL-105-I; AnshLabs, Webster, TX, USA) developed for use in humans and non-human primates. The assay has been applied successfully to assess blood AMH concentrations in previous studies of primates [6,7]. Prior to analysis, plasma samples were diluted 1:4 with sample diluent provided by the kit, and the assay was carried out according to the manufacturer’s instructions, with each sample measured in duplicate. Intra-assay coefficients of variation (CV) calculated from two high- and low-value quality controls run on the sample plate were 3.3% and 9.4%. As only one plate was run, no inter-assay CVs were determined.

### 2.6. Determination of Cortisol in Plasma Samples

Cortisol concentrations were determined from residual plasma blood samples, during the routine ovarian cycle monitoring, of 19 (STG: *n* = 12; LTG: *n* = 7) randomly selected animals frequently used as embryo donors (age: 49 ± 15 months). From each animal, blood samples were collected and analyzed for cortisol concentrations at three different time points, separated by 3 weeks each. All samples were collected during the luteal phase of the ovarian cycle. Cortisol was determined by an enzyme immunoassay using an antiserum against cortisol-3-CMO-BSA and biotinylated cortisol as enzyme conjugate [28]. Prior to analysis, plasma samples were diluted 1:10,000 in PBS buffer (pH 7.2) to establish cortisol concentrations within the working range of the assay. The assay was then performed as described in detail by [29]. Intra-assay CV values of high- and low-value quality controls were 7.5% and 8.8%, respectively, while inter-assay CVs were 8.9% and 9.9% [25]. The level of potential distress was further assessed based on the animals’ behavior during handling and blood sampling. Stress scoring was performed retrospectively by experienced animal caretakers who were familiar with the animals but blinded to the plasma cortisol levels. The subjective evaluation resulted in stress ratings from either moderate (+++), little (++), very little (+), or no (−) behavioral stress symptoms at all. Stress scores were later correlated with the cortisol levels determined in plasma.

### 2.7. Determination of Body Weight Changes

Changes in body weight following anesthesia were determined 1, 2, and 4 weeks after anesthesia in nine animals, including five animals from the STG and four from the LTG, for which sufficient weight data were available. Fluctuations in body weight are presented as percentage relative to the weight at time of anesthesia.

### 2.8. Statistics

The distribution of data was evaluated using descriptive statistics and Kolmogorov–Smirnov tests. Differences between two groups were analyzed using the Mann–Whitney U test for non-normally distributed independent samples or Student’s *t*-test for normally distributed samples, as appropriate (Figures 2b–d, 3b–d and 5d). Statistical evaluation of repeated measurements of blood parameters was performed using one-way ANOVA for repeated measures, followed by Tukey’s correction for multiple testing e.g., for the liver enzymes (Figure 1b), embryo retrieval data (Figures 2a and 3a), and cortisol data (Figure 5a–c,e). Body weight changes following anesthesia were calculated with mixed-model ANOVAs and Tukey’s correction for multiple testing. All analyses were performed using GraphPad Prism (Version 8), with *p*-values < 0.05 considered significant.

## 3. Results

### 3.1. Ovarian Cycles Remain Stable under Monthly Anesthesia

A stable and regular ovarian cycle in common marmosets is crucial for successful collection of embryos. We did not encounter cycle abnormalities of the embryo donors under experimental conditions, as judged from cycle length and hormone profiles. A representative progesterone profile of a naturally cycling female marmoset before and after the onset of monthly anesthesia is given in Figure 1a. Ovarian cycling remained stable in all animals despite monthly anesthesia and uterine flushes.

**Figure 1 animals-12-02414-f001:**
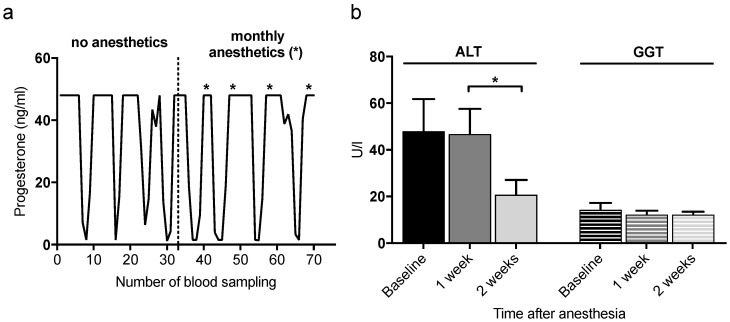
Repeated anesthesia neither impairs reproductive cycles nor hepatic health, as displayed by alanine aminotransferase (ALT) and gamma-glutamyl transferase (GGT) concentrations. (**a**) Representative progesterone profile of a marmoset before and during monthly anesthesia with ketamine, xylazine, and atropine. No general changes in the progesterone profiles were encountered after initiation of monthly anesthesia. (**b**) Liver enzymes ALT and GGT in serum samples before as well as 1 and 2 weeks after anesthesia as markers for hepatic health (*n* = 4). Both enzymes remained in the normal range of healthy marmosets at all time points. Bars represent mean ± SEM. * *p* < 0.05.

### 3.2. Liver Enzymes Do Not Increase upon Anesthesia

Anesthesia was usually achieved by injecting a mixture of ketamine and xylazine combined with atropine, and, for some anesthesias, also the combination of alphaxalone and diazepam. Elimination of ketamine and xylazine occurs via the liver pathway. Ketamine [30] has previously been reported to induce liver damage in human patients associated with an increase in the liver enzymes alanine aminotransferase (ALT) and gamma-glutamyl transferase (GGT). Both were measured in the blood of four randomly selected animals before anesthesia and 1 week and 2 weeks after anesthesia as a marker for hepatic health and function. At the time of blood sampling, animals had already undergone monthly cycles of anesthesia for embryo retrieval over a period of 1 to 3 years. There was no increase in either parameter after application of anesthesia compared to the respective baseline value before anesthesia (Figure 1b). ALT decreased from the second to the third time point significantly (*p* = 0.036), whereas only slight variations could be observed for GGT. As both liver parameters, ALT and GGT, remained within the physiological range at all time points tested [31], our data provide no evidence for hepatotoxic side effects caused by frequently repeated anesthesia.

### 3.3. ERRs in Different Subgroups

Several reports show a correlation between distress and reproductive outcome in female NHPs [32] and humans [33,34,35,36,37]. Here, we used the average number of naturally fertilized embryos retrieved per uterine flush [21,23], called embryo retrieval rate (ERR), as a marker for female fertility and reproductive health. Accordingly, the ERRs resulting from a total number of 667 uterine flushes performed on 32 animals were evaluated retrospectively.

The average ERR of all flushes of the 32 female marmosets was 0.781 (±0.985) embryos per flush. However, the animals were included in the studies for different periods of time and numbers of flushes (*n*) varied from 6–50 based on their subjective performance as embryo donors. Good embryo donors were used over a longer time period than poor embryo donors. Therefore, animals were divided into subgroups according to the total timespan used for uterine flushes. As shown in Figure 2a, and as expected from our pre-selection of animals according to their performance as embryo donors (see above), there was a gradual increase in the ERR depending on the animal group, with the lowest number of embryos per flush in animals that were used for 1 year or less for experiments, and the highest ERR in animals used up to 5 years for uterine flushes (*p* = 0.0005). These results confirm our subjective categorization of animals as good and poor embryo donors.


**Figure 2 animals-12-02414-f002:**
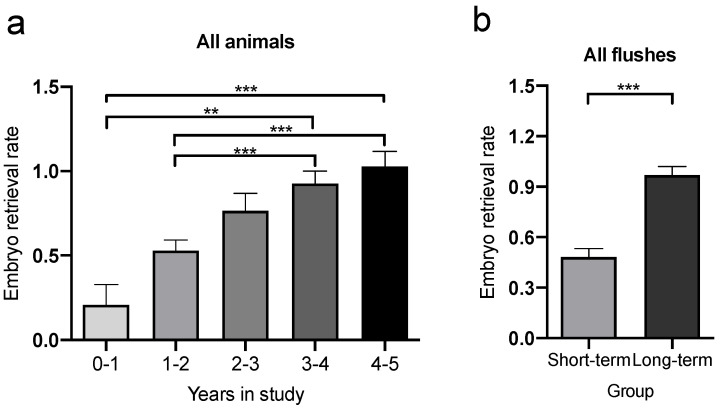
ERR in female marmosets (*n* = 32). (**a**) Cohorts of embryo donors included for different periods of time in the study show increasing ERR with extended inclusion periods. ** *p* < 0.01; *** *p* < 0.001 (**b**) Animals were separated into LTG and STG depending on the number of anesthesias and embryo flushes (LTG: ≥25 flushes; STG < 25 flushes). Marmosets assigned to the LTG (*n* = 12) display higher efficiency as embryo donors (*p* < 0.0001) throughout the study period compared to animals in the STG (*n* = 20). (**c**) High ERRs were already encountered during the first 2.5 years of experimental procedures (*p* < 0.0001) and (**d**) during the first year of study (*p* < 0.0001) compared to animals in the STG. Bars represent mean ± SEM. *** *p* < 0.001.

Based on these findings, we subdivided all animals into a long-term (LTG) and a short-term group (STG). This enabled us to examine whether poor embryo donors of the STG differed from those of the LTG in terms of distress and health parameters (see below). We used 2.5 years as a cutoff to distinguish between the two groups, as this time point represents the arithmetic mean of all tested animals. Moreover, the ERR of the group of animals that were used for 2–3 years for uterine flushes did not significantly differ from any other group. Accordingly, animals included for up to 2.5 years in the experimental procedures were assigned to the STG, whereas animals that were used between 2.5 and 5 years were assigned to the LTG. Mean and minimum age of animals in the long-term group was 37 and 15 months by study inclusion, whereas these values were 50 and 40 months for the short-term group. Since the groups were established based on their embryo donor efficiencies, it was not possible to obtain the same age for each group, as otherwise the number of animals in the groups would have had to be greatly decreased. Animals in the STG (*n* = 20) underwent up to 24 flushes each, resulting in a total number of 257 flushes. Animals in the LTG (*n* = 12) underwent up to 50 flushes each, with a cumulative number of 410 flushes. As can be seen from the data shown in Figure 2a, the ERR of the LTG (0.968 ± 1.052 embryos per flush) was significantly higher compared to that of the STG (0.482 ± 0.781) (*p* < 0.0001) (Figure 2b). Morphological quality and numbers of blastomeres of the retrieved embryos were checked, but no allocation to a quality group (e.g., good, fair, poor) was done since this information was not necessary in the three different studies the present data were obtained from. Therefore, a potential difference in embryo quality between the groups was not evaluated and hence not possible due to the retrospective character of the study.

To determine whether the higher ERRs in animals in the LTG could have resulted from habituation to study procedures over time, which may be missing in animals in the STG, thus biasing results, we compared the ERRs of animals in the LTG for the first 2.5 years used in experiments with the ERR of the same animals after 2.5 years. There was no significant difference in the early or late ERRs of the animals in the LTG (data not shown).

Next, we compared the ERR of animals in the STG with the ERR of the LTG in the corresponding timespan of 0–2.5 years. As shown in Figure 2c, animals in the LTG showed a significantly higher ERR (0.986 ± 1.053 embryos per flush) compared to animals from the STG (0.482 ± 0.781 embryos per flush; *p* < 0.0001) during the first 2.5 years. This prominent difference in the ERR between animals from the LTG and STG was already present in the first study year (*p* = 0.0002; 0.500 ± 0.828 vs. 0.907 ± 1.046 embryos per flush; Figure 2d), thus disproving the assumption that the higher ERR in the LTG may result from habituation over time.

Intrigued by the high average ERR over time, we next asked whether these high rates may be caused by a higher percentage of successful flushes, where at least one embryo could be retrieved. In line with the ERR, marmosets that were used only 1 year or less for uterine flushes showed the lowest percentage of successful flushes (9.8 ± 9.1%; Figure 3a). The percental amount of successful flushes showed a linear increase depending on the animal group, with highest proportion of successful flushes (58.7 ± 6.3%) in animals that were used up to 5 years for uterine flushes (*p* = 0.0023).

**Figure 3 animals-12-02414-f003:**
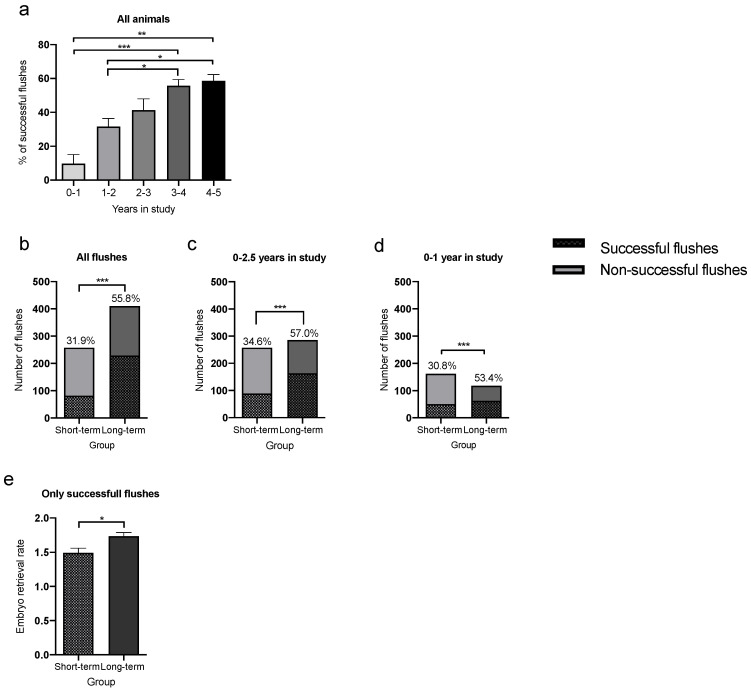
Differentiation between successful and unsuccessful flushes for embryo retrieval in female marmosets (*n* = 32). (**a**) Female marmosets show an increase in the percentage of successful flushes, that yielded at least one embryo. Animals in the LTG had a higher embryo flush success rate, with more than 50% of all flushes being successful (**b**) over the whole study period (*p* < 0.0001), (**c**) during the first 2.5 years within the study (*p* < 0.0001), and (**d**) during the first year of experiments (*p* = 0.0002). (**e**) When evaluating only the successful flushes, the ERR was higher in animals in the LTG (*p* = 0.0442) (*n* = 229 flushes) compared to animals in the STG (*n* = 81 flushes). Bars represent mean ± SEM. * *p* < 0.05; ** *p* < 0.01; *** *p* < 0.001.

In accordance with Figure 3b, LTG animals revealed a significantly higher percentage of successful flushes in terms of a positive outcome with at least one embryo (55.8% of all flushes combined; *p* < 0.0001) compared to the STG animals (31.9%; Figure 3b). If we only compare the percentage of successful flushes from the first 2.5 years of both groups, animals in the LTG also showed a significantly higher success rate compared to the STG (57% vs. 34.6%; *p* < 0.0001; Figure 3c). This pronounced effect was already detectable in the first study year (53.4% vs. 30.8%; *p* = 0.0002; Figure 3d). Hence, in LTG animals, every second uterine flush yielded at least one embryo, whereas in STG animals, only about every third flush resulted in one or more isolated embryos (Figure 3d).

Evaluation of only the successful flushes from both groups showed that LTG animals also provided a significantly higher number of embryos/flush (*p* = 0.0442). In summary, the percentage of successful flushes as well as the average number of embryos/successful flush was higher in the LTG compared to the STG.

### 3.4. AMH Concentrations in the LTG and the STG

Anti-Müllerian-hormone (AMH) is considered as a marker of the ovarian reserve and reflects the early follicular development. In our study, AMH concentrations were determined in randomly selected blood samples of 3-year-old animals either from the STG (*n* = 5) or the LTG (*n* = 5). As seen in Figure 4a, the mean AMH concentration in the STG was significantly lower (10.21 ± 6.63 ng/mL) compared to the LTG (20.21 ± 3.49 ng/mL). When comparing the ERRs from the tested animals, we detected a significantly lower number of retrieved embryos in the STG (0.46 ± 0.78) compared to the LTG (0.91 ± 1.04) (Figure 4b). Moreover, AMH concentrations displayed a positive correlation (r = 0.6445, *p* = 0.0443) with the ERR (Figure 4c), with low AMH concentrations in animals with a low ERR and high AMH values in animals with a high ERR.

### 3.5. Plasma Cortisol Concentrations over Time

Cortisol as the main glucocorticoid in most mammals reflects adrenocortical activity, and is often used as a measure of physiological distress in NHPs, including common marmosets [38]. In our study, the monthly applied anesthesia for embryo retrieval and the regular collection of blood samples from the inguinal vein up to six times per month may have been a cause of increased and cumulative adrenocortical activity reflecting a physiological stress response.

We analyzed blood cortisol levels in 19 randomly selected study animals at three different time points. The first sample (here called baseline) was taken within the first 2 weeks after inclusion into the study, followed by another two blood samples collected 3 and 6 weeks after the first sample. Blood cortisol levels decreased significantly from the first time point to the second (*p* = 0.0002) and third (*p* = 0.0064) time point (Figure 5a).

**Figure 5 animals-12-02414-f005:**
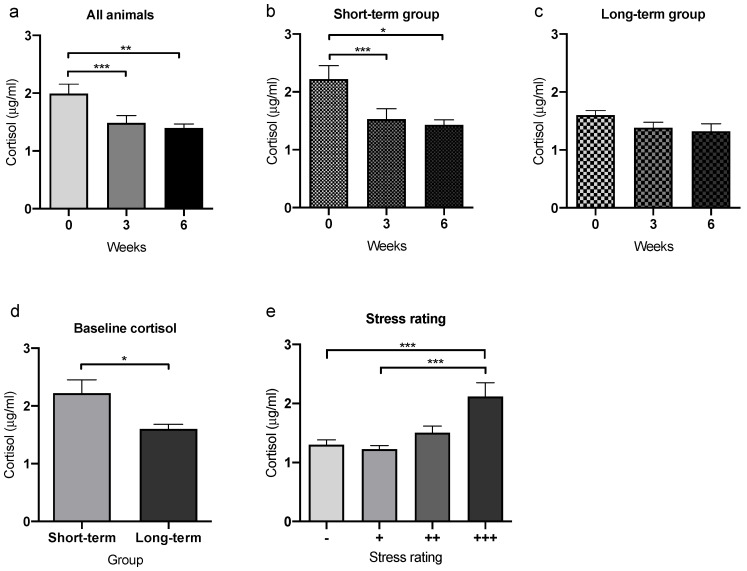
Quantification of blood cortisol concentrations and correlation with behavioral signs of distress. (**a**) Evaluation of cortisol concentrations in blood of female marmosets taken within the first 2 weeks after inclusion into the study as well as 3 and 6 weeks later. (**a**) Taking all tested animals together (*n* = 19), blood cortisol was highest shortly after study inclusion and declined significantly until 3 (*p* = 0.0002) and 6 (*p* = 0.0064) weeks later. When divided into the short-term (*n* = 12) and long-term (*n* = 7) group, only animals in the STG (**b**) showed significant declines in cortisol levels over the next 6 weeks. (**c**) In animals in the long-term group, only minor variations in cortisol levels were detectable, but (**d**) with already significantly lower levels at the first time point of cortisol measurements (called baseline) (*p* = 0.0121). (**e**) Correlation of blood cortisol levels with the behavioral stress rating, which was performed by the animal caretakers retrospectively without any information about laboratory parameters of the animals (*n* = 9 animals with scoring at 48 time points in total). Animals with the strong behavioral signs of stress had highest blood cortisol concentrations. Bars represent mean ± SEM. * *p* < 0.05; ** *p* < 0.01; *** *p* < 0.001.

We then analyzed the 19 animals based on their belonging to the STG or LTG. The decline in the cortisol levels were significant only in animals of the STG but not in those of the LTG. Animals in the STG (*n* = 12) showed a significant decline in blood cortisol from the first to the second (*p* < 0.0001) and from the first to the third time point (*p* = 0.0152; Figure 5b). Cortisol levels in the LTG (*n* = 7) showed only a modest non-significant decline (Figure 5c), mainly due to significantly lower cortisol levels already at the first time point compared to animals from the STG (*p* = 0.0121; Figure 5d).

To test whether using blood cortisol as a marker for physiological distress correlates with the animals’ individual appearance of being distressed during handling procedures, we asked the animal caretakers to retrospectively and subjectively assess the animals´ behavior during handling according to a simple rating system (−/+/++/+++, see Materials and Methods). All animal caretakers were blinded to serum cortisol values. Highest cortisol levels were found in animals with high behavioral stress ratings (+++; Figure 5e), which were significantly higher compared to animals displaying only minor (*p* = 0.0002) or no (*p* = 0.0002) signs of distress. Based on these results, we can conclude that there was an inverse correlation between ERR and initial serum cortisol levels. Furthermore, animals with initially high cortisol levels showed strong signs of distress in a subjective assessment performed by trained persons.

### 3.6. STG and LTG Animals Show Differential Body Weight Dynamics after Anesthesia

While humans are more likely to gain body weight under conditions of chronic distress [39], NHPs tend to put on less body weight or even lose weight when stressed or depressed [40]. Therefore, we evaluated the body weights 1, 2, and 4 weeks after each anesthesia as a marker for prolonged distress (Figure 6). Animals in the STG showed a pronounced decline in body weight 2 weeks after anesthesia (*p* = 0.0002 compared to time of anesthesia). This body weight loss was compensated only after another 2 weeks (*p* = 0.0008; Figure 6a). In contrast, LTG animals showed only a negligible decline in body weight 1 week after anesthesia and overcompensated this weight loss within 2 weeks (*p* = 0.001 compared to 1 week after anesthesia, Figure 5b). Furthermore, until the following anesthesia around 1 month after the preceding event, the body weight of the animals in the LTG had increased significantly (*p* = 0.001; Figure 6b), and were even more pronounced compared to 1 week after anesthesia (*p* < 0.0001, Figure 6b).

Overall, animals in the LTG showed a slight, yet non-significant decline in body weight 1 week after anesthesia, but a clear overcompensation of this mild decline in body weight during the subsequent weeks. In contrast, STG animals showed a significant decline in body weight of more than 2% at 2 weeks after anesthesia, which was compensated only after 4 weeks. Both study groups gained body weight during the course of the study (data not shown). Hence, body weight did not indicate significant long-term harm to the animals’ health and well-being, neither in the STG nor in the LTG; however, the profiles of the body weight fluctuations between both groups are different.

## 4. Discussion

To our knowledge, this is the first study analyzing effects of long-term experimental use of NHPs that includes frequently repeated anesthesia to examine parameters on their reproductive health and well-being. We determined the ERR as a stress-related parameter of fertility, liver enzymes, AMH concentrations, blood cortisol levels, and body weight changes in female common marmoset monkeys as physiological markers of distress. Although the repeated sessions of general anesthesia and uterine flushes did not compromise hepatic health or the animals’ health in general, our data indicate negative effects on animal well-being in the subgroup of short-term-used animals in terms of an expected lower ERR, higher initial cortisol levels and a transient decline in body weight. Remarkably, these effects were only seen in the STG, but not in the LTG, suggesting differential capabilities of animals to cope with distress associated with experimental procedures. Even more important, this difference between the two groups of animals could already be detected retrospectively in the first year of experiments, suggesting the existence of innate stress resilience in a subgroup of animals. These findings indicate that frequent and long-term handling and repeated anesthesia are not, per se, detrimental to the health and well-being of marmosets, and do not necessarily lead to stress accumulation over time, but indicate that the effects need to be considered for each animal individually.

Monthly anesthesia and regular blood sampling are potentially stressful procedures, and may affect steroid hormone levels [41]. Here, we used a combination of xylazine, ketamine, and atropine to induce anesthesia. Ketamine is an injectable anesthetic agent commonly used in NHP medicine [42] that is metabolized by the liver [43]. It can lead to hepatic inflammation, thereby elevating liver enzymes, such as ALT or GGT [44]. In all tested animals, the blood levels of ALT and GGT remained within the normal range [31] and did not increase over time. Thus, hepatotoxic side effects from repeated anesthesia represented by these two markers can be excluded at least for the tested animals. Unfortunately, liver enzyme analysis requires high blood volumes, which were only available for a very limited number of animals or time points.

Moreover, monthly anesthesia over extended periods of time, i.e., several years, did not alter the reproductive capacity of the animals in the LTG, as reflected by ERRs, or ovarian cycle activity of marmosets, as reflected by serum progesterone profiles. Our results are in line with a previous study on rhesus macaques that reported no effect of ketamine plus xylazine anesthesia on the profiles of reproductive hormones (i.e., estrogen, luteinizing hormone (LH), follicle-stimulating hormone (FSH)) or the occurrence of ovulation [45]. Nevertheless, for a deeper understanding of repeated anesthesia on steroidogenesis and folliculogenesis, further studies involving a narrower blood sampling regime, and the use of additional -omics technologies is needed.

A stable ovarian cycle with its two distinct phases, the follicular and the luteal phase, is crucial for reliable retrieval of naturally fertilized embryos. During distress, resources can be shifted from biological processes not relevant for immediate survival, such as reproduction, towards processes needed for immediate stress response and survival [46]. Here, we evaluated the effects of repeated and potentially stressful sessions of anesthesia, handling and blood sampling on fertility represented by the ERR and the embryo flush success rate (as defined by the rate of successful flushes of all flushes) for naturally fertilized embryos. The embryos were kept only for hours in the lab before being further processed in three different projects, therefore proper developmental competence of the embryos was not evaluated. Hence, only the quantity of embryos served as a biomarker for analysis of the embryo donor efficiency and the embryo quality was not separately monitored.

In our cohort of marmosets, some animals displayed higher ERRs compared to others. Interestingly, and against the assumption of accumulation of distress in animals undergoing long-term studies, the ERR did not decline over time. Rather, good embryo donors showed high ERR from the beginning until the end of the study, whereas animals with low ERR displayed low retrieval and low success rates already in the first study year without any increase in the ERR over time. Low performance in the ERR led to earlier exclusion from the study in order to avoid further manipulations without satisfying embryo outcome and additional harm to the animal. Importantly, the low ERRs were associated with higher cortisol levels and a pronounced transient post-surgery weight loss, suggesting that these animals were experiencing distress.

Of note, there is the possibility of bias introduction into our analysis since animals were allocated into STG and LTG according to the number of embryo retrievals received, which, in turn, depended on the subjective performance of the animals. However, in all animals, both the ERR and the embryo flush success rate remained stable from the beginning and throughout the animals’ inclusion in the study, which makes it very unlikely that the poor embryo donors would have improved later if not excluded early from the study. In our cohort of marmosets, high ERR in long-term-used animals does not seem to be the result of animal habituation to study procedures over time, but rather of an innate ability to better cope with potentially stressful situations. Moreover, it would be unethical and not justifiable to have animals with constantly low embryo numbers undergo additional experimental procedures with low chances of success. Additionally, the higher embryo retrieval and success rates in the LTG of animals, which remained at the same level throughout the study period, disproves the assumption that long-term exposure to potentially stressful study procedures, per se, negatively affects reproductive ability in marmosets.

Ovarian fitness can be quantified via the determination of AMH concentrations in blood samples. In humans, AMH concentration is a common marker to assess the ovarian reserve [5]. As AMH declines with age, we have chosen for reasons of comparability five animals at the age of 3 years from the STG and the LTG each for AMH measurement. As we found a positive correlation between AMH levels and embryo retrieval rates, we propose the use of AMH as a marker for the selection of animals for research studies involving high numbers of oocytes. Long et al. (2018) already showed in cynomolgus monkeys undergoing ovarian stimulation that AMH could serve as a cutoff marker to distinguish between animals with high or low ovarian reserve [6]. Nevertheless, animals included underwent hormonal ovarian stimulation, whereas in our study, naturally fertilized embryos were derived without prior hormonal stimulation of the animals. Furthermore, AMH has also been used in vervet monkeys by Atkins et al. (2014) previously. In summary, AMH seems to be a good marker for the ovarian reserve in marmosets as our data from this study and previous data suggest [8]. However, further evaluation of AMH concentrations in marmoset monkeys in relation to ovarian histology and under different biological and experimental conditions is required, and will be determined in future studies.

We further evaluated whether the lower ERR correlates with plasma levels of cortisol, a key biomarker for stress assessment. In response to distress, the hypothalamic-pituitary-adrenal axis becomes activated and releases increased amounts of cortisol to mobilize energy stores and prepare the body for action in order to respond to the stressor [46]. In diurnal animals such as the common marmoset, cortisol levels peak in the morning and decline throughout the day [47,48,49]. Therefore, all samples were collected at the same time in the morning and during the same phase of the ovarian cycle, even though blood cortisol levels have been reported to remain stable across the ovarian cycle of female marmosets and to be more dependent on the social status of the animal [49]. Cortisol levels varied substantially between individuals, which may result from different age and social status of tested animals, as described previously [50]. In our study, the individual differences in plasma cortisol correlated well with the stress rating assigned by the blinded animal caretakers, as animals perceived as being agitated or nervous during handling showed higher serum cortisol concentrations. However, regardless of these differences, in all tested animals the highest cortisol levels were detected within the first 2 weeks after inclusion into the study, when animals still needed to adapt to frequent handling. Regardless of their individual absolute cortisol concentrations, the cortisol levels declined significantly until week 6 of handling, and stabilized at this level until the end of the study (data not shown), suggesting that animals become used to the study procedures and do not experience long-term accumulation of distress.

In line with the results from embryo retrievals, animals in the STG showed significantly higher cortisol levels shortly after inclusion into the study. Although our data are based on a limited sample size, and thus need to be confirmed in future studies and interpreted with caution, it is conceivable that these higher cortisol levels may have directly contributed to the lower ERR found in animals of the STG, as long-lasting or chronic distress can downregulate processes non-essential to immediate survival, such as reproduction [51,52]. Moreover, glucocorticoid receptors are expressed in the ovary [53,54] and the uterus [55], exerting direct and local effects of cortisol on reproductive processes. In clinical studies, lower pregnancy rates following the application of ART have been associated with elevated level of cortisol [56,57]. Likewise, women undergoing ART, who reported higher infertility-related distress, anxiety, or depression, had lower treatment outcome and pregnancy rates [58,59], suggesting a direct link between psychological distress and reproduction in animals and humans.

In addition to plasma cortisol levels, distress may also be reflected by body weight changes. In 1985, Morton and Griffiths reported that body weight loss is a common clinical sign indicating pain, distress, or discomfort in laboratory animals, and can be measured objectively [60]. This also applies to NHPs, as for example depressed and stressed macaques show a strong reduction in body weight as a consequence of reduced food intake [61]. Another potential cause of reduced food intake and subsequently declining body weight is anesthesia with ketamine. Springer and Baker reported a significant reduction in daily food intake for 5 days after ketamine anesthesia before recovery to the baseline level [62]. Similar results were found for animals in the STG in the present study. These animals showed a significant loss of body weight 1 and 2 weeks after anesthesia, and compensated this weight loss only after approximately 4 weeks, shortly before the next anesthesia event. In contrast, animals in the LTG seemed to cope better with these potentially stressful situations, as they only displayed a minor and negligible weight loss 1 week after anesthesia, that was even over-compensated already after 2 weeks. A greater decline in body weight and slower compensation indicate a stronger impact of repeated sessions of anesthesia during embryo retrieval on well-being of animals in the STG compared to the LTG. Compromised well-being in these animals is further underlined by increased baseline cortisol levels, which may have synergistically led to the lower ERR. Nevertheless, the weight loss of animals in the STG was less than 5%, not posing any harm on the animals’ general health by itself.

The importance of animal welfare for reliable and predictive results derived from preclinical animal studies is increasingly recognized [63]. Only well-conducted studies taking animal welfare seriously into consideration will lead to reliable data that can serve as a solid basis for translational studies and development of novel medical therapies [63]. This is of particular importance for highly developed animals such as NHPs, which are sometimes used in experimental studies for several years, so that distress could accumulate over time. Especially for long-living NHPs, reuse of animals can offer many advantages and could lead to a reduction in animal numbers in compliance with the 3R-principle. Hence, it is necessary to develop objective criteria to evaluate not only acute pain and distress of animals, but also indicators of long-term cumulative burdens. This will contribute to both health and welfare of animals. Our data demonstrate that some animals cope better with repeated sessions of anesthesia, embryo retrieval, and blood sampling than other animals. Notably, this difference in stress coping was already seen after the first study year, equaling ten sessions of anesthesia. Hence, our data from the first study year obtained from marmosets have predictive power regarding the outcome in later years in terms of embryo retrieval. This is of importance for fertility studies or the generation of genetically modified NHPs, where high-quality embryos or oocytes in high numbers are needed. The well-being of embryo donor animals will increase the amount of naturally fertilized embryos, as seen in our LTG, and thus, may add to the study success. Moreover, it will also reduce the total number of animals needed to meet the study goals, emphasizing “reduction” of the 3R-principle. Hence, animals should be observed and pre-evaluated carefully, especially during the first weeks after inclusion into a study. This should be conducted not only for the sake of the animal, but also to ensure high quality and reliability of study results.

When conducting animal experimental studies, it is required to reduce the amount of invasively obtained samples such as blood (number of samples and volume) to what is necessary to achieve the purpose of the study. The current study was performed with samples that were available as residual samples of scientific work related to the isolation of preimplantation embryos for embryonic stem cell line derivation and the genetic modification of marmosets [13,21]. Therefore, often only small numbers of samples from animals from both groups (STG and LTG) were available to be meaningfully compared. This is a limitation of this study. However, it would be ethically questionable to conduct a stress study in NHP in which stressful and invasive procedures, including repeated anesthesia, were purposefully inflicted on animals just for the sake of analyzing stress. Therefore, despite its small sample sizes, we believe that this retrospective study can make an important contribution to improving 3Rs in primate research. Taken together, we provide evidence that some marmoset monkeys are less tolerant to frequent handling and study procedures such as repeated anesthesia and blood sampling than other animals. We have demonstrated that by a lower ERR, higher cortisol levels, more pronounced fluctuations of body weights following anesthesia as well as lower AMH concentrations in the STG compared to the LTG. As these parameters have been associated with reduced animal well-being during chronic distress, early and regular assessment of these markers may help to identify animals with a low stress tolerance early, limiting the (inefficient) repetition of study procedures in stress-sensitive animals thereby contributing to the principles of the 3Rs. In contrast, the data obtained from our LTG suggest that long-term use of stress-tolerant animals can also support the 3Rs in animal research by reducing the total number of animals required for certain experiments. In summary, to be in optimal compliance with the 3Rs and animal well-being, animals should ideally be individually tested for specific types of use, whenever possible.

## 5. Conclusions

Taken together, we provide evidence that some marmoset monkeys are less tolerant to frequent handling and study procedures, such as repeated anesthesia and blood sampling, compared to other animals. The animals from the STG used only up to 2.5 years seemed to be less stress-tolerant, as indicated by higher cortisol levels, more pronounced fluctuations in body weight following anesthesia, as well as lower AMH concentrations than animals used for more than 2.5 years. Since the tested markers serve as markers for distress and fertility, early and regular assessment of these markers may help to early identify animals with a low stress tolerance, limiting the (inefficient) repetition of study procedures in stress-sensitive animals, thereby contributing to the principles of the 3Rs. In contrast, the data obtained from our LTG suggest that long-term use of stress-tolerant animals can also support the 3Rs in animal research by reducing the total number of animals required for certain experiments. In summary, to be in optimal compliance with the 3Rs and animal well-being, animals should ideally be individually tested for specific types of use, whenever possible

## Figures and Tables

**Figure 4 animals-12-02414-f004:**
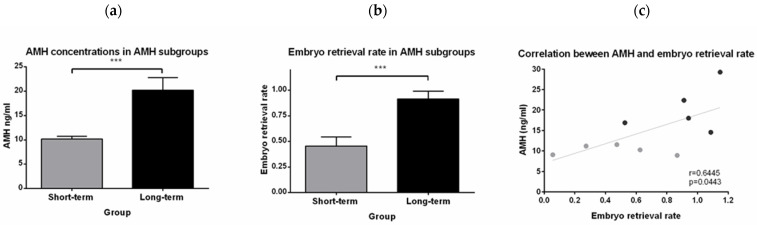
Determination of serum anti-Müllerian hormone (AMH) concentrations (in ng/mL) and correlation with ERR in subgroups. (**a**) Presentation of the AMH concentrations in ng/mL (*** *p* < 0.001) and (**b**) of the ERR (*** *p* < 0.001) in the STG (*n* = 5) and LTG (*n* = 5), respectively. (**c**) The ERR is positively correlated (*p* = 0.0443) with AMH concentrations. Animals in the STG (grey dots) cluster with low ERRs and low AMH concentrations and vice versa for animals in the LTG (black dots).

**Figure 6 animals-12-02414-f006:**
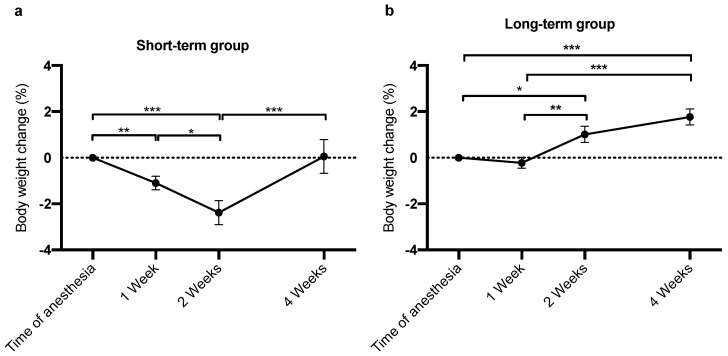
STG and LTG animals show different weight dynamics after anesthesia. (**a**) Animals in the STG (*n* = 5) displayed a significant decline in body weight 2 weeks after anesthesia compared to baseline (*p* = 0.0012), which was compensated only 4 weeks after anesthesia (*p* = 0.0012). (**b**) Body weight of animals in the LTG (*n* = 4) showed only negligible weight loss after anesthesia, which was fully compensated the following week and further increased until 4 weeks after anesthesia (*p* = 0.0004). Data are presented as mean ± SEM. * *p* < 0.05; ** *p* < 0.01; *** *p* < 0.001.

## Data Availability

Not applicable.

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
