# Peer review of "Performance of Marmoset Monkeys as Embryo Donors Is Reflected by Different Stress-Related Parameters"

_animals, 2022, doi:10.3390/ani12182414_

Round 1
Reviewer 1 Report
1. It would be helpful to see a table of demographic information for the 2 groups (STG,LTG) at baseline - age, AMH, body weight and social status. It would also be helpful to show these same data plus ERR data in animals with high cortisol compared to those with low cortisol.
2. It would be helpful to know more about housing of animals in the STG and LTG groups. For example were there STG animals paired with LTG animals?
Reviewer 2 Report
The authors evaluated the relationship between the performance of Marmoset monkeys as embryo donors and different parameters for chronic distress such as blood cortisol, liver enzymes and weight loss) in the manuscript. Therefore, they investigated whether repeated general anesthesia in the context of preimplantation embryo collection via uterine flushing and the associated frequent handling of female common marmosets (Callithrix jacchus) have negative long-term effects on health, well-being, and reproductive outcome. The subject of this study is suitable for “Animal” journal. The authors suggest that that long-term experimental use of marmosets does not impair fertility or health status. This is an interesting study and I suggest a few corrections to increase the scientific value of the manuscript. After these corrections were addressed by the authors, the manuscript can be accepted.
1. Experimental groups (long-term and short-term) should be clearly defined in terms of the number of animals, nutrition level, the days of use, body weight, etc. in the MM section.
1. How the relationship between the chronic distress stress parameters and long-term or short-term experimental use as a donor is determined should be explained in the statistical analysis section.
3. The results have been well presented and the results have been supported by the discussion, but the conclusion part should be arranged in a way that realistically emphasizes the results of the study.
Reviewer 3 Report
Non-human primates like marmoset monkeys are used as animal models for human diseases. To collect embryos, repeated anesthesia and frequent handling could result in chronic distress leading to sub- or even infertility. The aim of this manuscript is to retrospectively analyze the reproductive health of animals used as embryo donors employing measures of embryo retrieval rates (ERR), anti-Müllerian hormone (AMH) concentrations, cortisol levels and body weight fluctuations. Animals with successful embryo retrievals were grouped into a long- and short-term group (LTG and STG). Animals from the LTG group had higher and stable ERR associated with high AMH concentrations, low cortisol levels and minimal body weight fluctuations. The authors concluded that long-term use does not result in sub- or infertility supporting the reuse of animals.
The manuscript deals with the important question whether a reuse of animals is possible without affecting their health status and well-being. The paper is well-written in general. However, numerous issues need to be clarified/adjusted/complemented.
- The major concern goes to the point that only animals with successful embryo retrievals were included in the study. Do the authors know how many animals have been excluded as they did not deliver any embryo and do they know the possible reason(s)?
- The authors state that "AMH might be a marker …". Are there no data available for this species?
- The authors should mention that there might be other species which can be used as model for human diseases, at least for those in early development.
- How were the health and well-being of the animals monitored? Which parameters were examined?
- Two different procedures were used for embryo retrieval. The authors describe that both procedures showed the same efficiency and did not differ in their ERR or embryo flush success rates. Did the authors see differences in the other parameters?
- How efficient was the induction of luteolysis and reinitiation of a new ovarian cycle after Estrumate application?
- What is the number of animals being temporarily excluded from the experimental procedures (how often and how long)?
- At which age was the first embryo collection done in animals of both groups?
- What was the absolute number of liver enzyme analyses in animals of both groups?
- Did the authors check the quality of the embryos collected from animals of both groups?
- Is the degree of relationship known for the animals in both groups?
Round 2
Reviewer 3 Report
The authors have taken into account all of the points/issues raised by this reviewer. Most of them have been included in the manuscript. However, the author should include all points raised and all answers including the answers to questions 8, 10, and 11. Furthermore, the authors did not present data related to the quality of the collected embryos; f. ex. excellent, good, fair, poor. This needs to be included in the manuscript. They only described the developmental stage.
